# Meta-Learning in Self-Play Regret Minimization

## Abstract

Regret minimization is a general approach to on-line optimization which plays a crucial role in many algorithms for approximating Nash equilibria in two-player zero-sum games. The literature mainly focuses on solving individual games in isolation. However, in practice, players often encounter a distribution of similar but distinct games. For example, when trading correlated assets on the stock market, or when refining the strategy in subgames of a much larger game. Recently, offline meta-learning was used to accelerate one-sided equilibrium finding on such distributions. We build upon this, extending the framework to the more challenging *self-play* setting, which is the basis for most state-of-the-art equilibrium approximation algorithms for domains at scale. When selecting the strategy, our method uniquely integrates information across all decision states, promoting *global* communication as opposed to the traditional local regret decomposition. Empirical evaluation on normal-form games and river poker subgames shows our meta-learned algorithms considerably outperform other state-of-the-art regret minimization algorithms.

## 1. Introduction

Regret minimization has become a widely adapted approach for finding equilibria in imperfect-information games. Usually, each player is cast as an independent online learner. This learner interacts repeatedly with the game, which is represented by a black-box environment and encompasses the strategies of all other players or the game's inherent randomness. When all the learners employ a regret minimizer, their average strategy converges to a coarse correlated equilibrium (Hannan, 1957; Hart & Mas-Colell, 2000). Furthermore, in two-player zero-sum games, the average strategy

converges to a Nash equilibrium (Nisan et al., 2007). Regret minimization has become the key building block of many algorithms for finding Nash equilibria in two-player zero-sum imperfect-information games (Bowling et al., 2015; Moravčík et al., 2017; Brown & Sandholm, 2018; Brown et al., 2020; Brown & Sandholm, 2019a; Schmid et al., 2023).

While regret minimization algorithms have convergence guarantees regardless of the environment they interact with, they typically work significantly better in the *self-play* setting. In self-play, all players use a regret minimizer, rather than employing some adversarial strategy. The strategies used in self-play typically change much less than in an adversarial setting, resulting in smoother and faster empirical convergence to the equilibrium. This makes self-play a core component of many major recent successes in the field, which combine self-play and search with learned value functions (Moravčík et al., 2017; Schmid et al., 2023).

The literature on equilibrium finding mainly focuses on isolated games or their repeated play, with a few recent exceptions (Xu et al., 2022; Harris et al., 2022; Sychrovský et al., 2024). However, numerous real-world scenarios feature playing similar, but not identical games, such as playing poker with different public cards or trading correlated assets on the stock market. As these similar games feature similar equilibria, it is possible to further accelerate equilibrium finding (Harris et al., 2022).

An immediate application of a fast, domain-adapted regret minimization algorithm is online search (Moravčík et al., 2017; Brown & Sandholm, 2018; Schmid et al., 2023). In practice, an agent playing a game has limited time to make a decision. One thus seeks algorithms which can minimize regret quickly in the game tree. The computational time typically correlates well with the number of iterations, as evaluating the strategy at the leafs is often expensive. This is particularly true when a neural network is used to approximate the value function (Moravčík et al., 2017). Our approach allows one to trade time offline to meta-learn an algorithm which converges fast online.

Typically, established benchmarks are used to evaluate algorithms across a given field. Much of the literature about equilibrium finding focuses on finding algorithms with superior performance on these benchmarks. This is often done

[1] Anonymous Institution, Anonymous City, Anonymous Region, Anonymous Country. Correspondence to: Anonymous Author <anon.email@domain.com>.

Preliminary work. Under review by the International Conference on Machine Learning (ICML). Do not distribute.

by altering the previous state-of-the-art algorithms (Blackwell et al., 1956; Zinkevich et al., 2007; Tammelin, 2014; Farina et al., 2021; 2023). In this paper, we rather explore an adaptive approach, which finds a tailored algorithm for a domain at hand. Specifically, we use a variant of the 'learning not to regret' framework (Sychrovský et al., 2024), which we extend to the self-play setting. In this meta-learning framework, one learns the optimization algorithm itself. Our approach, in contrast to (Sychrovský et al., 2024), is a sound way to meta-learn regret minimizers in self-play, and significantly outperforms the previous state-of-the-art algorithms – on the very games they were designed to excel at.

### 1.1. Related Work

Meta-learning has a long history when used for optimization (Schmidhuber, 1992; 1993; Thrun & Thrun, 1996; Andrychowicz et al., 2016). This work rather considers meta-learning in the context of regret minimization. Many prior works explored modifications of regret matching (Blackwell et al., 1956) to speed-up its empirical performance in games, such as CFR$^+$ (Tammelin, 2014), LAZY-CFR (Zhou et al., 2020), DCFR (Brown & Sandholm, 2019b), LINEAR CFR (Brown et al., 2019), ECFR (Li et al., 2020), PCFR$^{(+)}$ (Farina et al., 2021), or SPCFR$^+$ (Farina et al., 2023).

It was recently shown that similar games have similar structures or even similar equilibria, justifying the use of meta-learning in games to accelerate equilibrium finding (Harris et al., 2022). A key difference between our and prior works is that they primarily consider settings where the game utilities come from a distribution, rather than sampling the games themselves. Thus, one of their requirements is that the strategy space itself must be the same. Azizi et al. (2023) consider bandits in Bayesian settings. Harris et al. (2022) "warm start" the initial strategies from a previous game, making the convergence provably faster. This approach is "path-dependent" in that it depends on which games were sampled in the past. Both works are fundamentally different from ours, as they use meta-learning online, while we are making meta-learning preparations offline, to be deployed online.

An offline meta-learning framework, called 'learning not to regret', was recently used to accelerate online play for a distribution of black-box environments encompassing two-player zero-sum games (Sychrovský et al., 2024). Their motivation, similar to ours, was to make the agent more efficient in online settings, where one has limited time to make a decision. In this setting, they wanted to minimize the time required to find a strategy with low one-sided exploitability, i.e. low distance to a Nash equilibrium.

### 1.2. Main Contribution

We extend the learning not to regret framework to the self-play setting (Sychrovský et al., 2024). First, we show that when using the prior meta-loss, the meta-learning will fail to convergence in the self-play setting. We formulate a new objective tailored to the self-play domain, which takes into account the strategies in the entire game. Our approach allow us to meta-learn regret minimizer which work well in an entire game, which enables their use in search. Search techniques are the basis for most state-of-the-art equilibrium approximation algorithms for domains at scale.

A unique feature of our method is we meta-learn the algorithms for both players and all the decision states simultaneously. We thus facilitate *global* inter-infostate communication. This is in contrast to the classic counterfactual regret decomposition, which works with *local* infostate regret and provides a bound on the overall regret (Zinkevich et al., 2007). We evaluate the algorithms on a distribution of normal-form and extensive-form two-player zero-sum games. We show that meta-learning can significantly improve performance in both settings.

## 2. Preliminaries

We briefly introduce two-player zero-sum incomplete information games. We then talk about regret minimization, a general online convex optimization framework. Finally, we discuss how regret minimization can be used to compute Nash equilibria of two-player zero-sum games.

### 2.1. Games

We work within a formalism based on factored-observation stochastic games (Kovařík et al., 2022).

**Definition 2.1.** A game is a tuple $\langle \mathcal{N}, \mathcal{W}, w^o, \mathcal{A}, \mathcal{T}, u, \mathcal{O} \rangle$, where

- $\mathcal{N} = \{1, 2\}$ is a **player set**. We use symbol $i$ for a player and $-i$ for its opponent.
- $\mathcal{W}$ is a set of **world states** and $w^0 \in \mathcal{W}$ is a designated initial world state.
- $\mathcal{A} = \mathcal{A}_1 \times \mathcal{A}_2$ is a space of **joint actions**. A world state with no legal actions is **terminal**. We use $\mathcal{Z}$ to denote the set of terminal world states.
- After taking a (legal) joint action $a$ at $w$, the **transition function** $\mathcal{T}$ determines the next world state $w'$, drawn from the probability distribution $\mathcal{T}(w, a) \in \Delta(\mathcal{W})$.
- $\mathcal{O} = (\mathcal{O}_1, \mathcal{O}_2)$ is the **observation function** which specifies the private and public information that $i$ receives upon the state transition.
- $u_i(z) = -u_{-i}(z)$ is the **utility** player $i$ receives when a terminal state $z \in \mathcal{Z}$ is reached.

The space $\mathcal{S}_i$ of all action-observation sequences can be viewed as the infostate tree of player $i$. A **strategy profile** is a tuple $\boldsymbol{\sigma} = (\boldsymbol{\sigma}_1, \boldsymbol{\sigma}_2)$, where each **strategy** $\boldsymbol{\sigma}_i : s_i \in \mathcal{S}_i \mapsto \boldsymbol{\sigma}_i(s_i) \in \Delta^{|\mathcal{A}_i(s_i)|}$ specifies the probability distribution from which player $i$ draws their next action conditional on having information $s_i$.

The **expected reward** (in the whole game) is $u_i(\boldsymbol{\sigma}) = \mathbb{E}_{z \sim \boldsymbol{\sigma}} \, u_i(z)$. The **best-response** to the other player's strategy $\boldsymbol{\sigma}_{-i}$ is $br(\boldsymbol{\sigma}_{-i}) \in \arg\max_{\boldsymbol{\sigma}_i} u_i(\boldsymbol{\sigma}_i, \boldsymbol{\sigma}_{-i})$. Finally, **exploitability** of a strategy $\boldsymbol{\sigma}$ is the sum of rewards each player can get by best-responding to his opponent

$$expl(\boldsymbol{\sigma}) = \frac{1}{|\mathcal{N}|} \sum_{i \in \mathcal{N}} u_i(br(\boldsymbol{\sigma}_{-i}), \boldsymbol{\sigma}_{-i}).$$

A strategy profile $\boldsymbol{\sigma}^*$ is a Nash equilibrium if it has zero exploitability.[1]

### 2.2. Regret Minimization and Nash Equilibra

An **online algorithm** $m$ for the regret minimization task repeatedly interacts with an **environment** through available actions $\mathcal{A}_i$. The goal of regret minimization algorithm is to maximize its hindsight performance (i.e., to minimize regret). For reasons discussed in the following section, we will describe the formalism from the point of view of player $i$ acting at an infostate $s \in \mathcal{S}_i$.

Formally, at each step $t \leq T$, the algorithm submits a **strategy** $\boldsymbol{\sigma}_i^t(s) \in \Delta^{|\mathcal{A}_i(s)|}$. Subsequently, it observes the expected **reward** $\boldsymbol{x}_i^t \in \mathbb{R}^{|\mathcal{A}_i(s)|}$ at the state $s$ for each of the actions from the environment, which depends on the strategy in the rest of the game. The difference in reward obtained under $\boldsymbol{\sigma}_i^t(s)$ and any fixed action strategy is called the **instantaneous regret** $\boldsymbol{r}_i(\boldsymbol{\sigma}^t, s) = \boldsymbol{x}_i^t(\boldsymbol{\sigma}^t) - \langle \boldsymbol{\sigma}_i^t(s), \boldsymbol{x}_i^t(\boldsymbol{\sigma}^t)\rangle \mathbf{1}$. The **cumulative regret** throughout time $t$ is $\boldsymbol{R}_i^t(s) = \sum_{\tau=1}^{t} \boldsymbol{r}_i(\boldsymbol{\sigma}^\tau, s)$.

The goal of a regret minimization algorithm is to ensure that the regret grows sublinearly for any sequence of rewards. One way to do that is for $m$ to select $\boldsymbol{\sigma}_i^{t+1}(s)$ proportionally to the positive parts of $\boldsymbol{R}_i^t(s)$, known as regret matching (Blackwell et al., 1956).

### 2.3. Connection Between Games and Regret Minimization

In two-player zero-sum games, if the **external regret** $R_i^{\text{ext},T} = \max_{a \in \mathcal{A}_i} \boldsymbol{R}_i^T$ grows as $\mathcal{O}(T^\delta)$, $\delta < 1$ for both players, then the average strategy $\overline{\boldsymbol{\sigma}}^T = \frac{1}{T} \sum_{t=1}^{T} \boldsymbol{\sigma}^t$ converges to a Nash equilibrium as $\mathcal{O}(T^{\delta-1})$ (Nisan et al., 2007).

In extensive-form games, in order to obtain the external

---

[1]This is because then the individual strategies are mutual best-responses.

---

**Algorithm 1** Neural Online Algorithm
(Sychrovský et al., 2024)

1: $\boldsymbol{R}^0 \leftarrow \mathbf{0} \in \mathbb{R}^{|A|}$
2: **function** NextStrategy($s$)
3:     $\boldsymbol{e}_s \leftarrow$ embedding of the infostate $s$
4:     $\boldsymbol{\sigma}^t \leftarrow m(\boldsymbol{r}^t, \boldsymbol{R}^t, \boldsymbol{e}_s \mid \theta)$
5: **function** ObserveReward($\boldsymbol{x}^t$)
6:     $\boldsymbol{R}^t \leftarrow \boldsymbol{R}^{t-1} + \boldsymbol{r}(\boldsymbol{\sigma}^t, \boldsymbol{x}^t)$

---

**Algorithm 2** Neural Predictive Regret Matching
(Sychrovský et al., 2024)

1: $\boldsymbol{R}^0 \leftarrow \mathbf{0} \in \mathbb{R}^{|A|}, \quad \boldsymbol{x}^0 \leftarrow \mathbf{0} \in \mathbb{R}^{|A|}$
2: **function** NEXTSTRATEGY()
3:     $\boldsymbol{\xi}^t \leftarrow [\boldsymbol{R}^{t-1} + \boldsymbol{p}^t]^+$
4:     **if** $\|\boldsymbol{\xi}^t\|_1 > 0$
5:        **return** $\boldsymbol{\sigma}^t \leftarrow \dfrac{\boldsymbol{\xi}^t}{\|\boldsymbol{\xi}^t\|_1}$
6:     **else**
7:        **return** $\boldsymbol{\sigma}^t \leftarrow$ arbitrary point in $\Delta^{|A|}$
8: **function** OBSERVEREWARD($\boldsymbol{x}^t, s$)
9:     $\boldsymbol{e}_s \leftarrow$ embedding of the infostate $s$
10:    $\boldsymbol{r}^t \leftarrow \boldsymbol{r}(\boldsymbol{\sigma}^t, \boldsymbol{x}^t)$
11:    $\boldsymbol{R}^t \leftarrow \boldsymbol{R}^{t-1} + \boldsymbol{r}^t$
12:    $\boldsymbol{p}^{t+1} \leftarrow \boldsymbol{r}^t + \pi(\boldsymbol{r}^t, \boldsymbol{R}^t, \boldsymbol{e}_s \mid \theta)$

---

regret, one would need to convert the game to normal-form. However, the size of this representation is exponential in the size extensive-form representation. Thankfully, one can upper-bound the normal-form regret by individual per-infostate **counterfactual regrets** (Zinkevich et al., 2007)

$$\sum_{i \in \mathcal{N}} R_i^{\text{ext},T} \leq \sum_{i \in \mathcal{N}} \sum_{s \in \mathcal{S}_i} \max\left\{ \|\boldsymbol{R}_i^T(s)\|_\infty, 0 \right\}. \quad (1)$$

The counterfactual regret is defined with respect to the **counterfactual reward**. At an infostate $s \in \mathcal{S}_i$, the counterfactual rewards measure the expected utility the player would obtain in the whole game when playing to reach $s$. In other words, it is the expected utility of $i$ at $s$, multiplied by the opponent's and chance's contribution to the probability of reaching $s$. We can treat each infostate as a separate environment, and minimize their counterfactual regrets independently. This approach again converges to a Nash equilibrium (Zinkevich et al., 2007).

## 3. Meta-Learning Framework

We aim to find an online algorithm $m_\theta$ with some parameterization $\theta$ that works "efficiently" on a distribution of regret minimization tasks $G$. However, what does it mean for a regret minimizer to be *good at minimizing regret on $G$*?

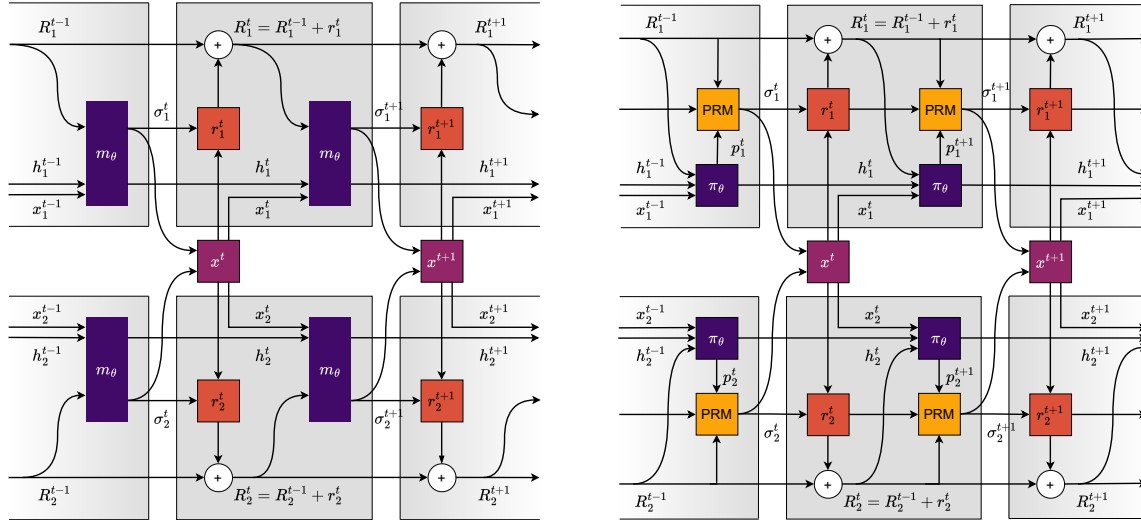

Figure 1: Computational graphs of NOA$^{(+)}$ (left) and NPCFR$^{(+)}$ (right). The gradient $\partial\mathcal{L}/\partial\theta$ originates in the collection of maximal instantaneous regrets $\left\|\boldsymbol{r}_i^{1\cdots T}\right\|_\infty$ and propagates through the strategies $\boldsymbol{\sigma}^{1\cdots T}$ (the predictions $\boldsymbol{p}_i^{1\cdots T}$ for NPCFR$^{(+)}$), the rewards $\boldsymbol{x}_i^{1\cdots T}(\boldsymbol{\sigma}^{1\cdots T}(\theta))$ coming from the rest of the game, the cumulative regret $\boldsymbol{R}_i^{0\cdots T-1}$, and hidden states $\boldsymbol{h}_i^{0\cdots T-1}$.

The simplest answer is to have the final external regret $R_i^{\text{ext},T}$ as small as possible for both players in expectation over $G$. However, minimizing it only forces the final average strategy $\overline{\boldsymbol{\sigma}}_\theta^T$ to be close to a Nash equilibrium.[2] Rather, we want the regret minimizer to choose strategies close to an equilibrium along *all the points of the trajectory* $\boldsymbol{\sigma}_\theta^1, \ldots \boldsymbol{\sigma}_\theta^T$, without putting too much emphasis on the horizon $T$. This is analogous to minimizing $\sum_{t=1}^T f(x^t)$ rather than $f(x^T)$ as in (Andrychowicz et al., 2016), where the authors meta-learned a function optimizer.

Consequently, we define the loss as the expectation over the maximum instantaneous counterfactual regret experienced at each step in all infostates of the game, i.e.

$$\mathcal{L}(\theta) = \mathop{\mathbb{E}}_{g\in G}\left[\sum_{i\in\mathcal{N}}\sum_{s_i\in\mathcal{S}_i(g)}\sum_{t=1}^T\left\|\boldsymbol{r}_i(\boldsymbol{\sigma}_\theta^t(s_i), \boldsymbol{x}^t(s_i|\theta))\right\|_\infty\right] \tag{2}$$

$$\geq \mathop{\mathbb{E}}_{g\in G}\left[\sum_{i\in\mathcal{N}}\sum_{s_i\in\mathcal{S}_i(g)}\left\|\boldsymbol{R}_i^T(s_i|\theta)\right\|_\infty\right] \geq \mathop{\mathbb{E}}_{g\in G}\left[\sum_{i\in\mathcal{N}}R_i^{\text{ext},T}(\theta)\right]$$

where the first inequality follows from convexity of the $\|\cdot\|_\infty$ norm, and the second from the full regret decomposition into individual counterfactual regrets (1).

Note that the loss (2) does not correspond to any kind of re-

_______________
[2]Empirically, the current strategy $\boldsymbol{\sigma}_\theta^t$ chosen by the regret minimizer remains quite far from the equilibrium and only the average strategy $\overline{\boldsymbol{\sigma}}_\theta^T$ approaches the Nash equilibrium when $\sum_{i\in\mathcal{N}}R_i^{\text{ext},T}$ is minimized.

gret that one can hope to minimize in an arbitrary black-box environment. However, it bounds the cumulative regret of both players, allowing us to indirectly minimize exploitability.

Another candidate for the meta-loss is the average external counterfactual regret along the trajectory

$$\mathop{\mathbb{E}}_{g\in G}\left[\sum_{i\in\mathcal{N}}\sum_{s_i\in\mathcal{S}_i(g)}\sum_{t=1}^T\left\|\boldsymbol{R}_i^t(s_i|\theta)\right\|_\infty\right]. \tag{3}$$

A telescopic argument shows that the minimum of both losses, assuming that $\theta$ has enough capacity, is the same. However, we found (3) harder to optimize, see Appendix A for further discussion.

The meta-loss (2) differs from the one used in (Sychrovský et al., 2024) in that the environment is not considered oblivious. This allows us to propagate the gradient thought the rewards $\boldsymbol{x}^t(s_i|\theta)$, and influence the opponent's strategy. In fact, when treating the environment as oblivious, (2) reduces to policy gradient (Sychrovský et al., 2024). The gradient would thus point towards the best-response to the current strategy of the opponent. While this approach does converge in an adversarial setting, such as when playing against a best-responder, it can cycle in self-play (Blackwell et al., 1956), see also Appendix A for further discussion.

Instead, our meta-loss takes the change in opponent's strategy into account. The difference can be observed even in a normal-form setting, where we get $\sum_{i\in\mathcal{N}}\|\boldsymbol{r}_i(\boldsymbol{\sigma}^t, \boldsymbol{x}^t)\|_\infty = \sum_{i\in\mathcal{N}}\|\boldsymbol{x}_i^t\|_\infty = \mathit{expl}(\boldsymbol{\sigma}^t)$. Thus,

minimizing (2) is equivalent to minimizing the expected exploitability of the selected strategy along the trajectory. Or in other words, $\boldsymbol{\sigma}_i^t$ minimizes the best-response value $\|\boldsymbol{x}_{-i}^t\|_\infty$ of the opponent. In extensive-form, the immediate counterfactual regret $\boldsymbol{r}_i(\boldsymbol{\sigma}_\theta^t(s_i), \boldsymbol{x}^t(s_i|\theta))$ reflects the structure of the game, and combines the strategies from different infostates in a non-trivial way.

We train a recurrent neural network $\theta$ to minimize (2). By utilizing a recurrent architecture we can also represent algorithms that are history and time dependent. Furthermore, this approach allows us to combine *all infostates of the game*. This is different from standard applications of regret minimization to games in which each infostate is optimized separately (Zinkevich et al., 2007). The local information strongly depends on the strategy selected at other infostates. In our approach, this can be sidestepped by directly accessing information from all infostates of the game tree, see Section 4 and Appendix C. To our best knowledge, our approach is the first to use cross-infostate communication in extensive-form games.

In the rest of this section, we briefly outline two meta-learning algorithms introduced in (Sychrovský et al., 2024). Their main difference is whether or not they enjoy regret minimization guarantees.

### 3.1. Neural Online Algorithm (NOA)

The simplest approach is to directly parameterize the online algorithm $m_\theta$ to output strategy $\boldsymbol{\sigma}_\theta^t$. This setup is refered to as neural online algorithm (NOA) (Sychrovský et al., 2024), see Algorithm 1. The computational graph of NOA is shown in Figure 1. While NOA can exhibit strong empirical performance on the domain it was trained on, there is no guarantee it will minimize regret, similar to policy gradient methods (Blackwell et al., 1956).

### 3.2. Neural Predictive Counterfactual Regret Minimization (NPCFR)

In order to provide convergence guarantees, (Sychrovský et al., 2024) introduced meta-learning within the predictive counterfactual regret minimization (PCFR) framework (Farina et al., 2021). PCFR is an extension of counterfactual regret minimization[3] (CFR) (Zinkevich et al., 2007) which uses an additional predictor about future regret. PCFR converges faster for more accurate predictions, and (crucially for us) enjoys $\mathcal{O}(1/\sqrt{T})$ convergence rate for arbitrary bounded predictions (Farina et al., 2021; Sychrovský et al., 2024). The neural predictive counterfactual regret minimization (NPCFR) is an extension of PCFR which uses a predictor $\pi$ parameterized by a neural network $\theta$, see

---

[3]Here we refer to using regret matching at each infostate rather than other regret minimization algorithm.

Algorithm 2. The resulting algorithm can be meta-learned to the domain of interest, while keeping the regret minimization guarantees. The computational graph of NPCFR is shown in Figure 1.

Predictive regret matching[4] was recently shown to be unstable in self-play when the cumulative regret is small. To remedy this problem, Farina et al. (2023) proposed the smooth predictive regret matching plus, which handles small cumulative regrets a special way. The resulting algorithm enjoys an $\mathcal{O}(1)$ bound on the external regret in self-play on normal-form games. Moreover, it can be meta-learned in the same way as NPCFR. However, we found the algorithm after meta-learning the predictor empirically performs nearly identically to NPCFR, see Appendix D.1 for more details.

## 4. Experiments

We focus on the application of regret minimization in games. We study two distributions of two-player zero-sum games, see Appendix B for their detailed description. First, we modify a normal-form game by adding a random perturbation to some of the utilities. For evaluation in extensive-form, we use the river subgames of Texas Hold'em poker. For all meta-learned algorithms, the neural network is based on a two-layer LSTM and uses all infostates of the game to produce $\boldsymbol{\sigma}_\theta^t$, see Appendix C for details.

In addition to the last-observed instantaneous regret $\boldsymbol{r}^t$ and the cumulative regret $\boldsymbol{R}^t$, the networks also receives an encoding of the infostate $\boldsymbol{e}_s$ and keeps track of its hidden state $\boldsymbol{h}^t$. The network is able to internally combine information from all infostates of the game, allowing for global communication between different infostates, see Appendix C. Finally, we define NOA$^+$ and NPCFR$^+$ using the 'plus' modifications[5] introduced in (Tammelin, 2014), and train them analogously.

We minimize objective (2) for $T = 32$ iterations over 256 epochs using the Adam optimizer. Other hyperparameters were found via a grid search. To prevent overfitting, we employ entropy regularization which decays inversely with the number of epochs. The meta-training can be completed in ten minutes on a single CPU for the normal-form, and ten hours for the extensive-form experiments.

For evaluation, we compute exploitability of the strategies up to $2T$ iterations to see whether our algorithms can generalize outside of the horizon $T$ and continue reducing exploitability. Both regimes use the self-play setting, i.e. each algorithm controls strategies of both players. We compare

---

[4]More precisely its 'plus' version (Tammelin, 2014).

[5]That includes (i) removing negative parts of the regret, (ii) alternating updates, and (iii) linear averaging.

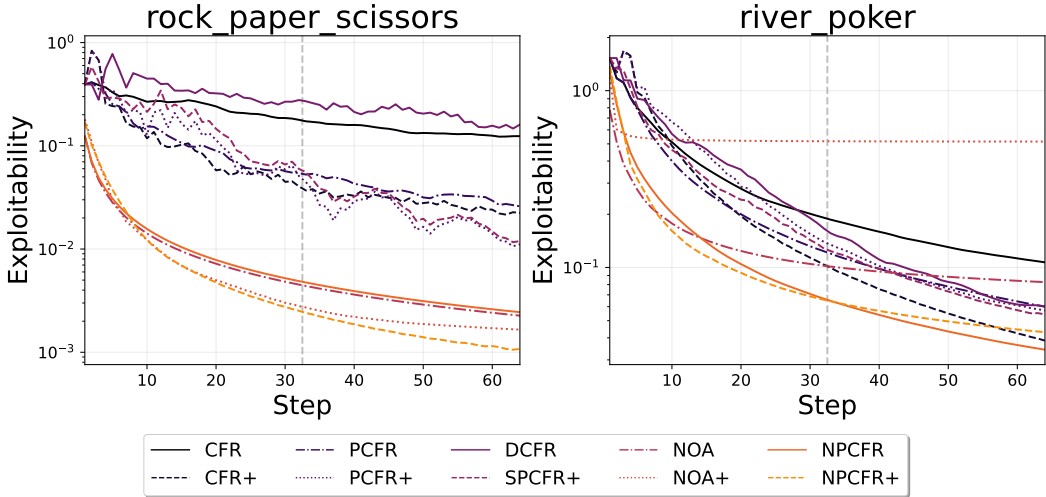

Figure 2: Comparison of non-meta-learned algorithms (CFR$^{(+)}$, PCFR$^{(+)}$, DCFR, and SPCFR$^+$) with meta-learned algorithms (NOA$^{(+)}$ and NPCFR$^{(+)}$) on `rock_paper_scissors` (left) and `river_poker` (right). The figures show exploitability of the average strategy $\overline{\sigma}^t$. Vertical dashed lines separate the training (up to $T = 32$ steps) and the generalization (from $T$ to $2T$ steps) regimes. See Figure 6 for standard errors.

our meta-learned algorithms with an array of state-of-the-art regret minimization algorithms. In particular, we use CFR$^{(+)}$ (Blackwell et al., 1956; Tammelin et al., 2015), PCFR$^{(+)}$ (Farina et al., 2021), DCFR[6] (Brown & Sandholm, 2019b), and SPCFR$^+$ (Farina et al., 2023).

### 4.1. Normal-Form Games

For evaluation in the normal-form setting, we use a modification of the standard `rock_paper_scissors`, perturbing two of its elements. Figure 2 shows our algorithms converge very fast. In fact, all the meta-learned algorithms outperform their non-meta-learned counterpart throughout time $T$, often by an order of magnitude. It continues to hold this advantage beyond what it was trained for. The 'plus' meta-learned algorithms show better performance.

To further illustrate their differences, we plot the trajectory of the average strategy $\overline{\sigma}^t$ selected by each algorithm in Figure 3. The meta-learned algorithms exhibit much smoother convergence, while choosing strategies which are inside to the region of equilibria that are possible in `rock_paper_scissors`.[7] In contrast, the non-meta-learned algorithms visit large portions of the strategy-space. We analogously show the current strategy $\sigma^t$ in Figure 5, Appendix D.

### 4.2. Extensive-Form Games

We use Texas Hold'em poker to evaluate our algorithms in the sequential decision making. Specifically, we use the river subgame, which begins after the last public card is revealed. We fix the stack of each player to nine-times blind, which translates to $24,696$ infostates.[8] To generate a distribution, we sample beliefs over cards at the root of the subgame as in (Moravčík et al., 2017). We refer to the resulting distribution as `river_poker`. The meta-learned algorithms can utilize additional contextual information about the infostate it is operating at. In our case, this encoding includes two-hot encoding of the private cards, five-hot encoding of the public cards, and the beliefs over cards in the root.

We compare the exploitability of the strategy found by the meta-learned algorithms to their non-meta-learned counterparts in Figure 2, see also Figure 7 in Appendix D for the comparison of external regret. The meta-learned algorithms exhibit superior performance, which manifests itself in the reduced number of iterations needed to achieve a given solution quality, see Table 1. While all meta-learned algorithms exhibit fast convergence initially, the generalization of NOA$^{(+)}$ is considerably worse than those of NPCFR$^{(+)}$. We hypothesize that, since NOA$^{(+)}$ are not regret minimizers, the strategy in a subset of infostates can remain poor,

---

[6]We use the default parameters $\alpha = 3/2, \beta = 0$, and $\gamma = 2$ (Brown & Sandholm, 2019b).

[7]Thanks to the alternating updates, the strategy of the 'plus' versions of each algorithm is uniform at $t = 1$.

[8]We use the fold, check, pot-bet, and all-in (FCPA) betting abstraction (Gilpin et al., 2008).

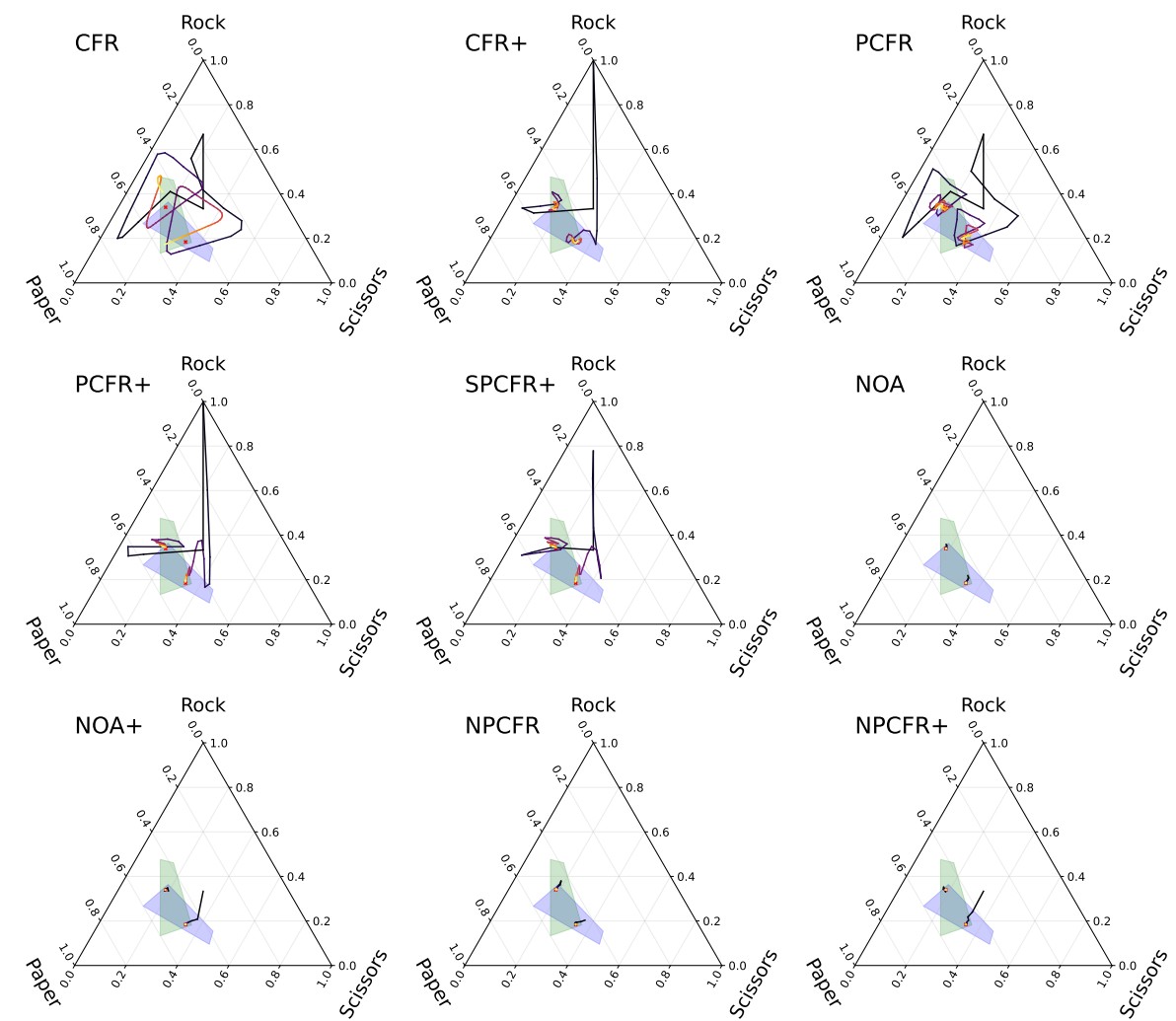

Figure 3: Comparison of the convergence in average strategy $\overline{\boldsymbol{\sigma}}^t$ on a sample of `rock_paper_scissors` over $2T = 64$ steps. The red crosses show the per-player equilibria of the sampled game. The quadrilaterals show the region of equilibria (Bok & Hladík, 2015). We use blue for the first player and green for the second. The trajectories start in dark colors and get brighter for later steps. See Figure 5 in Appendix D for current strategy convergence.

hindering the exploitability in the whole game.

As was observed before, the 'plus' version of the non-meta-learned algorithms tend to perform better on this domain (Tammelin, 2014). Surprisingly, this is not the case for the meta-learned algorithms, in particular for NOA⁺. We believe that the discrete operations introduced by the 'plus' interfere with the gradient of the meta-loss, causing the training to be unstable.

The PCFR algorithm guarantees that more accurate predictions lead to lower external regret and faster convergence. We can thus ask if the meta-learned algorithms use such accurate predictions. Surprisingly, this turns out not to be the case, see Figure 8 in Appendix D. We speculate that the reason is that regret matching is not an injective mapping.

Indeed, any prediction which results in $[\boldsymbol{R}^{t-1} + \boldsymbol{p}^t]$ being proportional will lead to the same $\boldsymbol{\sigma}^t$ being selected.

### 4.2.1. COMPUTATION TIME REDUCTION

Using a neural network in the meta-learned algorithms introduces a non-trivial amount of computational overhead. In our implementation, when using a single CPU, the non-meta-learned algorithms require roughly 10 ms to complete a regret minimization step in the tree of `river_poker`. In contrast, the meta-learned algorithms are about three-times slower. Therefore, despite requiring less iterations to reach given exploitability, the computation using the meta-learned algorithms is more expensive.

This is because, in the 'full' subgame, we are able to com-

| $expl$ | CFR | CFR$^+$ | PCFR | PCFR$^+$ | DCFR | SPCFR$^+$ | NOA | NOA$^+$ | NPCFR | NPCFR$^+$ |
|---|---|---|---|---|---|---|---|---|---|---|
| 1 | 4 | 5 | 4 | 7 | 5 | 5 | **1** | **1** | 2 | 2 |
| 0.5 | 11 | 10 | 9 | 14 | 14 | 10 | **3** | $> 512$ | 4 | 4 |
| 0.1 | 70 | 33 | 40 | 41 | 44 | 40 | 35 | $> 512$ | 21 | **19** |
| 0.05 | 162 | 54 | 77 | 71 | 73 | 71 | $> 512$ | $> 512$ | **44** | 50 |

Table 1: The number of steps each algorithm needs to reach a given exploitability threshold on `river_poker` in expectation.

pute the terminal utilities relatively fast as they are just hand strength comparisons. However, this is no longer the case when doing search, where the utility at the leaves of the truncated subgame tree are typically neural network approximations of the remaining value of the game. Typically, the value function is represented by a large neural network, introducing computational overhead (Moravčík et al., 2017). Depending on the network, this could be $10 - 100$ ms even when using specialized hardware accelerators, making our approach which significantly reduces the number of iterations required much more competitive. We show several examples in Figure 12 in Appendix D. Finally, looking beyond poker, each terminal evaluation required for the regret minimization step can be expensive in other ways. For example, it can involve obtaining the terminal values involves querying humans (Ouyang et al., 2024), or running expensive simulations (Degrave et al., 2022).

**4.3. Out-of-Distribution Convergence**

To reinforce the fact that the meta-learned algorithms are tailored to the training domain $G$, we evaluate the algorithms out-of-distribution, see Figure 9 in Appendix D. We use the regret minimizers trained on `rock_paper_scissors`, see Section 4.1. We freeze the network parameters $\theta$ and evaluate them on the `uniform_matrix_game`, see Appendix B. The performance of all meta-learned algorithms deteriorates significantly. NOA$^{(+)}$ shows the worst performance, seemingly failing to converge all together. NPCFR$^{(+)}$ are regret minimizers and are thus guaranteed to converge, albeit slower than the general-purpose non-meta-learned algorithms.

**5. Conclusion**

We've extended the 'learning not to regret' (Sychrovský et al., 2024), a meta-learning regret minimization framework, to the self-play setting of two-player zero-sum games. We evaluated the meta-learned algorithms and compared them to state-of-the-art on normal-form, and extensive-form two-player zero-sum games. The meta-learned algorithms considerably outperformed the prior algorithms in terms of the number of regret minimization steps. These gains can be particularly impactful in search, where each regret mini-

mization step includes evaluating a value function, which typically dominates the computational cost.

**Future Work.** We find that, in extensive-form, there is typically a large gap between our meta-loss and the exploitability of the strategy. Exploring other losses which tighten this gap could improve performance. While the meta-learning scales well, it can be impractical when applied to larger games or longer horizons. Training on an abstraction of a particular game of interest might open the door to scaling it further. This approach also allows one to compare different abstractions, and uncover important features of a class of games. Our framework can also be directly used within other algorithms utilizing search. In particular within the Student of Games (Schmid et al., 2023), in place of CFR$^+$ to drastically reduce the number of subgame interactions needed for resolving on the subgame of interest.

**Impact Statement**

This paper presents work whose goal is to advance the field of Machine Learning. There are many potential societal consequences of our work, none which we feel must be specifically highlighted here.

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
