# OpenReview forum: "Meta-Learning in Self-Play Regret Minimization"
_ICML.cc/2025/Conference — Submitted to ICML 2025_

### Official Review · Reviewer_CExE · 2025-02-16

**Overall Recommendation:** 2

**Summary:**

Traditional self-play methods are often used to compute equilibria in large, extensive-form, two-player, zero-sum games. This submission studies meta-learning in games in the self-play setting, motivated by the observation that many real real world decision making problems involves a distribution of related-but-distinct games (e.g. financial trading, poker sub-games). Previous work on meta-learning in games focuses on optimizing regret minimization algorithms for one-sided equilibrium finding in a sequence of games. This work attempts to generalize this idea to self-play, where both players adapt their strategies simultaneously.

The authors build on the “learning not to regret” framework of Sychrovsky et al. (2024), but identify that the original formulation may fail to converge in self-play due to cycling behavior. To address this, they identify a new meta-loss function designed to prevent cycles and ensure stable learning. Unlike traditional regret minimization methods like CFR, which update “local” regrets at each decision state independently, the authors’ method introduces global communication across decision states. They evaluate their approach on normal-form games and river poker subgames, and compare their performance to state-of-the-art regret minimization algorithms.

**Claims And Evidence:**

Yes.

**Essential References Not Discussed:**

No.

**Experimental Designs Or Analyses:**

Yes.

**Methods And Evaluation Criteria:**

Yes.

**Other Comments Or Suggestions:**

n/a

**Other Strengths And Weaknesses:**

While the authors’ meta-learned algorithms do reduce the number of iterations required for convergence, they come with notable drawbacks.

Empirically, the choice of using neural networks increases the computational overhead of their method, compared to other alternatives. Furthermore, on the river poker subgames, a non-meta-learning method achives very similar performance to the authors’ proposed method. From my understanding, the per iteration runtime of the authors’ method is higher, and so it is unclear whether this approach offers any net-improvement over non-meta-learned techniques.

To summarize, this paper presents an ambitious but ultimately limited extension of meta-learning to self-play regret minimization. While the introduction of global communication across decision states is an interesting departure from classical regret minimization techniques, the inconsistent empirical performance weaken its impact. In theory, meta-learning should accelerate convergence in self-play games, but it appears that the high computational cost makes the results of this work less competitive against the well-established CFR-based baselines.

**Questions For Authors:**

n/a

**Relation To Broader Scientific Literature:**

This submission builds off of the small-but-quickly-growing literature on meta-learning in games. Specifically, they target meta-learning in the self-play setting.

**Theoretical Claims:**

There are no theoretical claims in this submission.

---

> ### Author Rebuttal · Authors · 2025-03-31
>
> We would like to sincerely thank the reviewers for their time spent to help improve our work.
> We appreciate all the comments and will integrate them into a revised version of the paper.
> Let us address the questions and comments raised.
>
> We would like to politely disagree with your point about the lack of convergence guarantees of our algorithms.
> In fact, the whole point of our paper is how to introduce meta-learning to regret minimization without losing the convergence guarantees.
> This is explained in Section 3.2 of our paper, with detailed references to relevant earlier work.
>
> The per-iteration cost of our approach is indeed higher.
> Studying the trade-off between the improved per-iteration convergence and the inference speed is a major part of our experimental section.
> In Section 4.2.1, we show that our algorithms outperform the prior ones even when taking this into account.

---

> > ### Comment · Reviewer_CExE · 2025-04-01
> >
> > Perhaps I am missing something. PCFR enjoys convergence guarantees, but the neural version does not, no?

---

> > > ### Author Response · Authors · 2025-04-04
> > >
> > > The neural version of PCFR does enjoy no regret guarantees. This is because PCFR enjoys guarantees regardless of the prediction. This was shown in Sychrovsky 2024, Thm 1.

---

### Official Review · Reviewer_4NiF · 2025-02-25

**Overall Recommendation:** 1

**Summary:**

This paper extends meta-learning (i.e., learning a regret minimizer algorithm over a sequence of games drawn from a distribution) to the self-play setting. In particular, it derives a new meta loss for training a regret minimizer. The performances of this procedure are evaluated on two-player normal form and extensive form zero-sum games.

**Claims And Evidence:**

The main claim of the paper is that the new meta-loss allows global inter-infostate communication, leading to better performances in practise. This claim is not supported by any theoretical results. Instead, the authors perform two experiments, one in on a rock-paper-scissor example (normal form game) and another one on Texas Hold'em poker (extensive-form game) where they compare their approaches to standard non-meta-learned algorithms. I find it unfortunate that the authors have not tried to derive theoretical guarantees on the performance of their algorithms. In the absence of theoretical results, I would expect much more experiments for the above mentioned claims to be credible.

**Essential References Not Discussed:**

I do not think about any essential reference that has not been discussed.

**Experimental Designs Or Analyses:**

The experiments are well detailed and explained, which I appreciated.

**Methods And Evaluation Criteria:**

I am not convinced that the two experiments are enough to show the superiority of the proposed approach. In particular, CFR+ seems to perform as well, if not better, than the proposed algorithms on river poker (figure 2) for >60 steps. Confidence intervals from repeated experiments are definitely lacking here, to check whether one procedure significantly (in a statistical sense) outperforms another. Additionally, it would have been preferable to run the experiments over more steps to get insights about the asymptotic behaviors of the algorithms. T=32 seems a bit short here.

**Other Comments Or Suggestions:**

I have no other suggestions.

**Other Strengths And Weaknesses:**

STRENGTHS:
- The experiments are nice and well described (in particular the poker example), although they are not ambitious enough from my point of view.

WEAKNESSES:
- I do not clearly see what the contribution of this paper is, in particular as compared to [1].
- The claims of the paper are not well supported. While I have nothing against fully empirical papers (which this paper is), two experiments over T=32 timesteps are not convincing.
- The paper is not really well written. Some key concepts (such as as self-play) are barely defined, some parts are useless (e.g. definition 2.1, which is never used again the rest of the paper!), some mathematical objects are not well defined (e.g. the "hidden states" h_t, line 251).  I find the main text not precise enough and sometimes hard to understand.

Overall, I do not think that the paper in its current form matches the requirements of the conference.

**Questions For Authors:**

Q1. Can you clarify what are the contributions of this paper as compared to [1]?

Q2. It seems that the loss defined in (2) (line 203) features the regret of both players. In other words, it seems that you are training a centralized model which optimizes both players' strategies. This contradicts the game-theoretic framework, wohse whole point is exactly that players take actions in an uncoupled way. In particular, it makes no sense to speak about Nash equilibrium if strategies are correlated through a central model. Can you clarify this point ?

**Relation To Broader Scientific Literature:**

I do not clearly understand the contribution of this paper, in particular as compared to [1]. Indeed, the authors seem to use the same algorithms as in [1] (with a different loss), and the rock-paper-scissor experiment is identical to the one in [1]. The authors claim that the present paper is an extension of [1] to the "self-play" setting, however they never clearly state what this means. The only description of what "self-play" is supposed to mean in this article is the following:

"In self-play, all players use a regret minimizer, rather than employing some adversarial strategy."

I do not see why the fact that other players use a regret minimizer is a novelty. On the contrary, it is the most common assumption in the literature of learning in games. This point needs to be clarified.

[1]. Sychrovský, D., Šustr, M., Davoodi, E., Bowling, M., Lanctot, M., & Schmid, M. (2024, March). Learning not to regret. In Proceedings of the AAAI Conference on Artificial Intelligence (Vol. 38, No. 14, pp. 15202-15210).

**Theoretical Claims:**

There is no theoretical claim in this paper.

---

> ### Author Rebuttal · Authors · 2025-03-31
>
> We would like to sincerely thank the reviewers for their time spent to help improve our work.
> We appreciate all the comments and will integrate them into a revised version of the paper.
> Let us address the questions and comments raised.
>
> We don't want to claim the cross-infostate communication guarantees better empirical performance.
> Our algorithm is simply the first of its kind which allows for it, which we highlight in the paper.
>
> Other algorithms, such as CFR+, do indeed outperform our algorithms, but only outside of their training domain.
> In practice, one would thus need to train the algorithm for a sufficient number of steps.
> However, even outside of the training domain, the algorithms keep minimizing regret at a steady pace.
> We include the figures with standard errors in the Appendix.
>
> Our paper extends the work of Sychrovsky 2024 to the self-play domain.
> When doing self-play, regret minimization algorithms empirically converge much faster.
> In fact, self-play enabled many past successes in the field, including DeepStack and Student of Games.
> However, as mentioned in the paper, the work of Sychrovsky 2024 cannot be applied in this setting, as the meta-learning problem they formulate is not well posed.
> In our work, we extend their work, which enables its use in practical applications.
>
> Regarding you second question, we politely disagree.
> All algorithms for finding Nash equilibria need to do so by finding a strategy for both players.
> Even the simplest algorithms, such as support enumeration, internally work with strategies of both players.
> As you pointed out, self-play is a widely used framework, which by definition needs to consider strategies of both players at the same time.
> Algorithms such as Smooth predictive regret matching+ even guarantee faster convergence rates because they work with both players' strategies.
> In the end of the day, finding Nash equilibria in two-player zero-sum games is an optimization problem, which can be solved by considering both players at the same time.

---

> > ### Comment · Reviewer_4NiF · 2025-04-03
> >
> > I appreciate the authors' response. I acknowledge that the proposed algorithm is the first to enable cross-infostate communication. I also understand that optimizing over both players' strategies is reasonable in a self-play setting—although, as I mentioned in my review, this is not clearly conveyed in the paper.
> >
> > However, the authors have not addressed my concerns regarding the limited experiments, the lack of theoretical analysis, and the paper's writing quality. Consequently, I am not changing my recommendation.

---

### Official Review · Reviewer_svcG · 2025-03-11

**Overall Recommendation:** 3

**Summary:**

The authors extend Neural Online Algorithm (NOA) and Neural Predictive Regret Matching (NPRM) from Sychrovský et al., 2024 to the self-play setting, creating a meta-learned self-play regret minimizer. This is done by modeling the computational graphs for both players instead of just one. In two-player zero-sum games, they demonstrate faster exploitability reduction within two domains of games compared to non-meta-learned regret minimization techniques.

**Claims And Evidence:**

The claims are mainly supported by evaluation on two distributions of games, modified normal-form rock paper scissors, and extensive form river subgames from Texas Hold'em poker. Evaluating on only two games is quite limited.

**Essential References Not Discussed:**

No essential missing references that I can discern.

**Ethical Review Concerns:**

No ethics concerns.

**Experimental Designs Or Analyses:**

One weakness with the evaluation is that a gridsearch was performed on key parameters for the proposed method, but not for the baselines.

**Methods And Evaluation Criteria:**

The games considered are appropriate. The proposed evaluation criteria makes sense. They train on a distribution of games each for 32 optimization steps and evaluate on the same distribution for 64 optimization steps. Exploitability vs regret minimization steps is the appropriate evaluation metric for the two-player zero-sum game setting. I was also glad to see experiments on out-of-distribution games.

**Other Comments Or Suggestions:**

Many of the appendix figures look similar. Adding titles to the appendix figures would make it easier to discern which is which.

**Other Strengths And Weaknesses:**

Strengths:
- Extending meta-learning regret minimization to self-play tackles an important problem, regardless of the scale that this algorithm operates at.
- Writing is clear, other than the network training procedure.

Weaknesses:
- Lacking in empirical validation, only evaluating in two small game domains.
- Hyperparameters for baselines could have also been tuned to the tested domains.
- Unclear description of the network architecture optimization and training loop (see questions).
- Depending on how exactly the network is optimized, I have scalability concerns. If scalability is a problem, to address this weakness, it needs to be acknowledged more thoroughly than it currently is.

**Questions For Authors:**

Could the authors please clarify how the max pooling is performed, and what a batch update looks like in this optimization process? The exact nature of the global inter-infostate communication is unclear to me. It sounds like the max-pooling is performed to aggregate values over every infostate in the game. At an optimization level, is the aggregation actually done over minibatches or a trajectories, etc, or is it truly over every single game infostate? If so, the authors should add clearer wording about the performance and scalability limitations of doing so.

**Relation To Broader Scientific Literature:**

This paper places itself sufficiently well within the broader subfield of regret minimization in games. They clearly specify distinctions and contributions between this work and "Learning Not to Regret" (Sychrovský et al., 2024) by adapting the method to the case where both players are jointly optimized with the meta-learned regret-minimization algorithm.

**Theoretical Claims:**

I did not verify the correctness of the proofs.

---

> ### Author Rebuttal · Authors · 2025-03-31
>
> We would like to sincerely thank the reviewers for their time spent to help improve our work.
> We appreciate all the comments and will integrate them into a revised version of the paper.
> Let us address the questions and comments raised.
>
> While the normal-form games we evaluate our algorithms can be viewed as toy examples, the River poker is a standard large benchmark. We consider all 54 choose 5 = 3,162,510 games in this distribution, each with 24,696 infostates.
> This is a rich class of games, which provides a good testbed for evaluating algorithms' performance.
>
> In terms of hyperparameter selection, our work differs from standard applications of deep learning in that we have the entire distribution and simply want to fit it as well as possible.
> Thus, we cannot gain much by selecting hyperparamters based on the test set, as it is our training set as well.
>
> The network architecture we use is described in the main text, and in more details in Appendix C.
> The max pooling layer was selected because it makes the network parameters constant in the game size -- the best scaling one can hope for in our case.
> It is used to aggregate the hidden state of the first LSTM layer across all infostates every time the network is called.
> It is done in independently for each instance, so communication between iterations or batch elements is not possible with this layer.
>
> Note that using the max pooling is not vital, and one can use other network architectures to allow for infostate communication.
> For example, replacing the max pooling with a fully-connected layer would make the network parameters scale with the square on the number of infostates.
>
> When scaling the game, because we use the same parameters for each infostate, the network size stays constant.
> One can increase the size of the layers and improve performance, at the cost of iteration speed.
> The exact tradeoffs depend on the game considered, and we discuss them in our experiments.
> We see no reason why our approach should be harder to scale than any other deep learning system.

---

> > ### Comment · Reviewer_svcG · 2025-04-09
> >
> > Thanks for the explanation. I would suggest going into more detail, spelling out how the model is trained to help readers, since it is so structurally different from other MARL approaches like NFSP, PSRO, MMD, etc.
> >
> > Overall, I maintain my current score recommendation.

---

### Decision · Program_Chairs · 2025-05-01

**Decision:**

Reject

**Comment:**

The reviewers largely agree -- the paper has potential value.  However, they suggest improvements are needed with respect to results and clarity of presentation.  As such, the paper is not ready for publication in ICML at this time.